# Beyond Turing: Topological Closure as a Foundation for Cognitive Computation

## Abstract

Classical models of computation, epitomized by the Turing machine, are grounded in *enumeration*: syntactic manipulation of discrete symbols according to formal rules. While powerful, such systems are intrinsically vulnerable to Gödelian incompleteness and Turing undecidability, since truth and meaning are sought through potentially endless symbolic rewriting. We propose an alternative foundation for non-enumerative computation based on *topological closure* of semantic structures. In this view, cognition operates by promoting transient fragments into closed cycles, where $\partial^2 = 0$ ensures that only invariants persist. This shift reframes computation from *syntax* to *structure*: memory and reasoning arise not by enumerating all possibilities, but by stabilizing relational invariants that survive perturbations and generalize across contexts. We formalize this principle through the dot–cycle dichotomy: dots or trivial cycles ($H_0$) serve as high-entropy scaffolds for exploration, while nontrivial cycles ($H_1$ and higher) encode low-entropy invariants that persist as memory. Extending this perspective, we show how Memory-Amortized Inference (MAI) implements an anti-enumerative principle by storing homological equivalence classes rather than symbolic traces, yielding robust generalization, energy efficiency, and structural completeness beyond Turing-style models. We conclude that *topological closure* provides a unifying framework for perception, memory, and action, and a candidate foundation for cognitive computation that transcends the limits of enumeration.

## 1 Introduction

Since the early 20th century, formal models of computation have been grounded in *enumeration*. The Turing machine, along with its close relatives in the Church–Turing framework, exemplifies this paradigm: computation is conceived as the syntactic manipulation of discrete symbols on an infinite tape, with new results obtained only through the stepwise application of formal rules Turing (1936). This model has proven enormously successful, forming the foundation of digital computing, automata theory, and modern complexity classes. Deep learning architectures, despite their connectionist implementation, inherit this enumerative character: generalization emerges by statistical interpolation over enumerated training examples, and inference requires repeated evaluation across contexts Goodfellow et al. (2016). Yet the power of enumerative systems is matched by their intrinsic limitations. Gödel's incompleteness theorem demonstrates that no sufficiently expressive formal system can be both complete and consistent: there will always exist true statements that cannot be proven within the system. Turing's halting problem further establishes that no algorithm can decide, in finite time, whether arbitrary programs will terminate Sipser (1996). Both results reveal a deeper structural fragility: *enumeration can never guarantee closure*. Each attempt to list or decide the totality of possible outcomes leaves residual boundaries, open fragments that escape formal capture. This fragility manifests as brittleness in symbolic AI, combinatorial explosion in search Minsky (1961), and distributional failures in data-driven models. Enumeration fails because it cannot stabilize residual structures, open chains proliferate without ever closing. Topological closure reframes this failure: what enumeration leaves dangling, closure promotes into invariants.

In this paper, we propose an information topological framework for intelligence in which *cycle closure* is the fundamental mechanism of memory. Building on the first principle, we argue that memory is best understood not as a static store of representations, but as the ability to *re-enter and traverse latent cycles* in neural state space. We identify these invariant cycles as the natural carriers

of meaning across scales: they act as *alignment checkpoints* between context ($\Psi$) and content ($\Phi$), filtering out order-specific noise, enforcing closure, and preserving only what remains consistent across variations. A key principle underlying this framework is the *dot-cycle dichotomy*: trivial cycles collapse to dots ($H_0$), serving as transient contextual scaffolds ($\Psi$), while nontrivial cycles ($H_1$ and higher) encode low-entropy content invariants ($\Phi$) that persist as memory. This dichotomy clarifies how cognition achieves both adaptability and stability: dots support exploration, while cycles carry persistent knowledge across contexts. From this perspective, cognition is not tape-based symbol manipulation but the promotion of transient fragments into closed cycles that survive perturbation and generalize across contexts. This shift from *syntax* to *structure* reframes memory, learning, and reasoning as processes of stabilizing invariants, not enumerating sequences. Under this new conceptual framework, we develop the following arguments:

- We explore the physical origin of intelligence inspired by the first clue in Wheeler (2018): $\partial^2 = 0 \ \Rightarrow \ $ Cycles (invariants) $\ \Rightarrow \ $ Memory $\ \Rightarrow \ $ Prediction (intelligence).

- We introduce the *dot-cycle dichotomy*: dots ($H_0$) encode disconnected fragments, while cycles ($H_1$ and higher) represent nontrivial order invariants that persist as memory.

- We introduce *Structure-before-Specificiy* principle as the guidance of memory organization. Structural content is represented by low-entropy homology and specific context serves as high-entropy scaffolding.

- We show how *Memory-Amortized Inference (MAI)* implements a context-content uncertainty principle (CCUP) by bootstrapping and retrieval operators, yielding energy efficiency and robust generalization.

## 2    Motivation: Intelligence as Topological Closure

A unifying way to interpret both Gödel's incompleteness theorem and Turing's halting problem is to see them as demonstrations of the failure of *countable closure*. Any attempt to exhaustively enumerate truths or procedures inevitably leaves a residue, a diagonal element, an undecidable program, that lies outside the reach of the list. From a topological perspective, this means that enumerations generate fragments that remain open boundaries, unable to close into global invariants. What escapes enumeration is not accidental but principled: closure requires invariants beyond counting. This reinterpretation shifts the focus from the fragility of syntactic lists to the robustness of semantic cycles. The implication is profound: if Wheeler's dictum It-from-Bit Wheeler (2018) highlights the informational substrate of reality, then for intelligence the relevant unit is not the fleeting bit but the persistent *cycle* that survives across variations Davatolhagh et al. (2024). Formally, we have

**Principle 1** (First Principle of Intelligence). *Intelligence is the capacity to stabilize invariants by cycle closure. At its core, cognition operates by minimizing joint context-content uncertainty $H(\Psi, \Phi)$, eliminating dangling boundaries and promoting them into closed cycles. These cycles constitute the fundamental units of meaning, memory, and prediction.*

Our guiding claim is that *cycle is all you need*: the organization of cognition, memory, and abstract thoughts in neural systems follows from the universal role of cycles as the algebraic residue of broken symmetry and the topological skeleton of information flow. This claim is supported by the hierarchical organization of cycles in mammalian brains, such as *Theta–gamma nesting* (e.g., hippocampus–entorhinal Buzsáki (1996)) and perception-action cycles Fuster (2004). In the spirit of Wheeler Wheeler (2018), we propose the following four No's for cognition.

1. **No isolated information.** Bits are never standalone: they acquire meaning only through relations that close into cycles. Information without recurrence dissipates as noise.

2. **No privileged order.** The cognitive system must be robust to permutations of local steps. What matters is closure into a cycle, not the linear order of micro-events.

3. **No specificity before structure.** Persistent structures must stabilize first as the backbone of memory and prediction, while contextual specificities become scaffolding later to provide adaptive flexibility.

4. **No prediction without invariance.** Forecasting future states requires reducing entropy by filtering order-dependent variations; only invariant cycles can stabilize the predictive substrate.

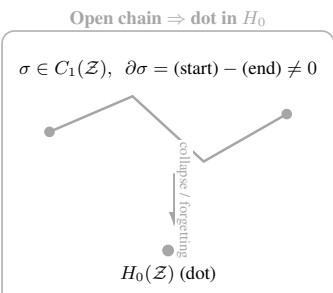
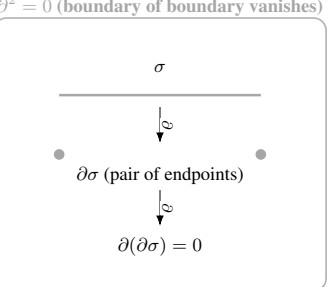
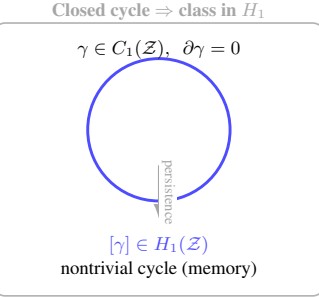

Figure 1: $\partial^2 = 0$ **enforces the dot-cycle dichotomy.** *Left:* An open chain $\sigma$ has a nonzero boundary $\partial\sigma$ and collapses to a dot (class in $H_0$), carrying no relational content. *Middle:* The boundary operator squares to zero: $\partial(\partial\sigma) = 0$. *Right:* A closed chain $\gamma$ with $\partial\gamma = 0$ persists as a homology class $[\gamma] \in H_1$, i.e., a cycle that encodes order-invariant structure.

**From constraints to clues.** These four principles define cognition as a non-ergodic information process Walters (2000): rather than averaging over all possible trajectories, the mind concentrates its dynamics onto recurrent, invariant cycles that persist across perturbations. Taken together, the Four No's funnel cognition toward recurrent organization: items must close into cycles (no isolated information), be insensitive to micro-order (no privileged order), support re-entry (no static storage), and stabilize invariants for prediction (no prediction without invariance). The lightest formalism that enforces all four at once is the chain complex with boundary operator $\partial$ Hatcher (2002): its nilpotency, $\partial^2 = 0$, cancels stray endpoints so that only closed traversals remain. This is the key new insight underlying the dot-cycle dichotomy, as shown in Fig. 1, and it sets up our first clue.

**Theorem 1** (The Boundary of a Boundary Vanishes). *Under the First Principle, intelligence is realized through cycle closure. This closure is only possible because the boundary operator $\partial$ satisfies the fundamental identity $\partial^2 = 0$. That is, the boundary of a boundary vanishes. Cognitively, this law ensures that when cognition promotes boundaries into cycles, no further inconsistencies remain at the next level: every open edge is paired, every fragment canceled. This guarantees the existence of stable invariants (cycles), which are the carriers of meaning, memory, and communication. Therefore, $\partial^2 = 0$ constitutes the* First Clue *of intelligence: coherence arises because boundaries consistently vanish when lifted, enabling cycles to persist.*

The vanishing of boundaries guarantees that what remains in memory is not arbitrary fragments but coherent cycles: the minimal invariants that bind context and content into an intelligible whole. From a computational standpoint, this marks a profound departure from the Turing paradigm. Traditional machines rely on symbolic tokens and sequential operations, where meaning is assigned externally to states of a register or tape. In contrast, a cycle-based architecture derives meaning intrinsically from topological closure: invariants are not "written" into memory but emerge from the very dynamics of neural interaction Gerstner et al. (2014). This dot–cycle dichotomy, *where trivial cycles collapse and only nontrivial cycles persist*, provides a natural mechanism for error correction, generalization, and energy efficiency without requiring exhaustive symbolic manipulation or gradient descent over high-dimensional parameter spaces. Rather than preserving measure by averaging over all paths, intelligent systems learn to *concentrate* probability mass onto order-invariant cycles (i.e., cycle-preserving structure replaces measure-preserving flow).

**Example 1** (Toy Navigation Loop). *In a $5 \times 5$ grid with a square obstacle (a "hole"), trajectories that poke the obstacle and backtrack are open $1$-chains ($\partial\sigma \neq 0$) and collapse to trivial $H_0$ "dots." By contrast, any homing route that circles the hole and returns to start yields a closed $1$-chain $\gamma$ with $\partial\gamma = 0$ and $[\gamma] \neq 0$ in $H_1$. Crucially, reordering the same edges (e.g., north-first vs. east-first) produces the* same *class $[\gamma]$: the loop is order-invariant and reusable as a navigation template.*

## 3 MEMORY AS STRUCTURED TRAJECTORIES IN THE LATENT SPACE

Classical ergodic theory is built on the notion of a measure-preserving transformation Walters (2000). A dynamical system $(X, \mathcal{B}, \mu, T)$ consists of a probability space $(X, \mathcal{B}, \mu)$ and a measurable transformation $T : X \to X$ satisfying $\mu(T^{-1}A) = \mu(A), \quad \forall A \in \mathcal{B}$. This measure invariance guarantees that long-term time averages along almost every trajectory coincide with ensemble averages

with respect to $\mu$. In this setting, entropy (e.g., Kolmogorov-Sinai entropy Cornfeld et al. (2012)) quantifies the unpredictability of the evolution under the assumption of ergodicity. Intelligent systems, however, are fundamentally non-ergodic: they retain memory, exhibit path dependence, and actively reduce uncertainty. In such systems, the measure $\mu$ is not preserved, but typically *concentrated* onto lower-dimensional recurrent structures through learning and adaptation Spisak & Friston (2025). This concentration corresponds to entropy minimization rather than entropy conservation.

We propose that the appropriate generalization of "measure-preservation" in the non-ergodic setting is *cycle-preservation*. That is, while probability measures are not conserved globally, the system preserves *topological invariants* encoded in cycles that represent memory traces and recurrent behavioral motifs Gromov (1999). Formally, let $(X, T)$ be a discrete-time dynamical system on a topological state space $X$. A $k$-cycle is a chain $\gamma \in Z_k(X)$ satisfying $\partial\gamma = 0$. Under the induced map $T_*$ on chains, invariance of $\gamma$ requires that $T_*\gamma - \gamma = \partial\beta$ for some $(k + 1)$-chain $\beta$. Equivalently, $[T_*\gamma] = [\gamma]$ in $H_k(X)$, where $H_k(X)$ denotes the k-th homology group of the topological space $X$, so that $\gamma$ is invariant up to homology class. In this way, although trajectories deform under dynamics (e.g., refer to the example of Wilson-Cowan model below), the *memory* encoded by the homology class persists.

**Example 2** (Wilson-Cowan Model). *The Wilson-Cowan system Wilson & Cowan (1972) $\dot{E} = -E + S(w_{ee}E - w_{ei}I + P)$, $\dot{I} = -I + S(w_{ie}E - w_{ii}I + Q)$ (with sigmoidal $S$) undergoes a supercritical Hopf bifurcation for an open set of parameters, yielding a hyperbolic limit cycle $\Gamma$. Under small bounded input/parameter perturbations, trajectories* deform *(phase/amplitude modulation) but structural stability preserves a nearby periodic orbit $\Gamma_\varepsilon$; thus the cycle, and its homology class $[\Gamma_\varepsilon] \in H_1$,* persists *even as paths vary.*

This shift in perspective reframes the role of entropy reduction. In ergodic systems, entropy is managed by distributing trajectories uniformly across the entire state space $X$, ensuring statistical equivalence of time and ensemble averages. By contrast, in non-ergodic, adaptive systems, entropy reduction is achieved through *measure concentration* Gorban & Tyukin (2018): rather than exploring all of $X$, trajectories are funneled toward lower-dimensional recurrent sets. These recurrent sets correspond to *persistent cycles* that remain stable under perturbations and across variations in initial conditions. In this sense, cycles act as the carriers of invariant information, preserving structural regularities across history-dependent dynamics and filtering out order-specific noise. The outcome is that intelligence emerges not from uniform exploration, but from the ability to stabilize information flow through the persistence of these invariant structures Ayzenberg et al. (2025). Formally, we have

**Principle 2** (Non-Ergodic Invariance Principle). *Let $(X, T)$ be a dynamical system on a topological state space $X$. Then the natural counterpart of measure-preservation in ergodic theory is* cycle-preservation: $T_* : H_k(X) \to H_k(X), [\gamma] \mapsto [\gamma]$. *That is, an intelligent system preserves homology classes of cycles even while its measure evolves non-uniformly. These invariant cycles formalize memory persistence as the structural backbone of cognition.*

When a non-ergodic system with many symmetric possibilities is forced to choose one outcome, symmetry is broken Anderson (1972). In neural and cognitive dynamics, this choice does not erase the unselected alternatives; instead, it organizes them into a closed cycle of relations: the chosen state, its competitors, and the transitions among them. In other words, the brain does not simply "pick a winner" among symmetric options. It establishes a cycle that records the selection, keeps the alternatives accessible for recall or switching, and stabilizes the outcome through recurrent interaction Hochreiter & Schmidhuber (1997). Broken symmetry, therefore, inevitably produces cycle formation, since the invariant residue of selection is a cycle connecting choice, memory, and potential revision.

This perspective reframes the role of entropy in prediction. Principle 2 establishes that non-ergodic systems preserve homology classes of cycles as their structural invariants. From an information-theoretic viewpoint, symmetry corresponds to maximal uncertainty: if all outcomes are equivalent under a symmetry group $G$, the induced distribution is uniform (entropy is maximized). Symmetry breaking reduces this uncertainty by eliminating redundant possibilities, thereby lowering entropy and concentrating probability mass around residual invariant cycles. In high dimensions, this process can be understood through the *theory of measure concentration* Ledoux (2001): instead of spreading trajectories uniformly, the dynamics of learning and memory focus trajectories around persistent cycles. To make this precise, we introduce the notion of *residual invariants* Beekman et al. (2019):

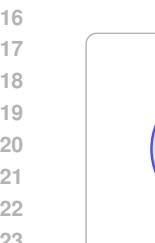 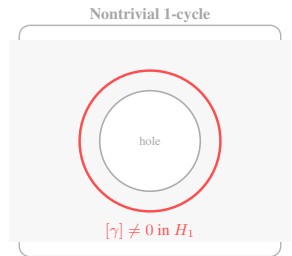 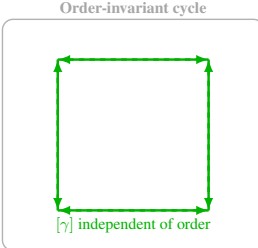

**Figure 2: Trivial, nontrivial, and order-invariant cycles.** *Left:* A boundary of a filled region is trivial in $H_1$. *Middle:* A loop around a hole cannot bound any 2-chain, so it represents a nontrivial homology class. *Right:* Once a trajectory closes into a cycle, its homology class depends only on the multiset of moves, not their order: order permutations yield the same $H_1$ class.

the structural survivors of symmetry breaking concentrate probability mass onto persistent cycles and formalize what remains stable under the reduced symmetry subgroup.

**Definition 1** (Residual Invariants under Symmetry Breaking)**.** *Let a system evolve on a state space $\mathcal{Z}$ with symmetry group $G$. Suppose a perturbation $\varepsilon$ breaks $G$-equivariance, reducing the symmetry to a subgroup $H \subset G$ and forcing selection of a representative state $\Phi_\varepsilon \in \mathcal{Z}$. The* residual invariants *are those structures that remain preserved under $H$ despite the breaking of $G$. Formally, they are equivalence classes of cycles $[\gamma] \in H_k(\mathcal{Z})$ that are stable under $H$-action and persist under perturbations of $\varepsilon$.*

Intuitively, residual invariants encode what remains stable after a decision or perturbation: in physics, they correspond to conserved quantities or Goldstone modes Beekman et al. (2019); in topology, to persistent homology classes Edelsbrunner et al. (2008); and in cognition, to cycles that bind chosen outcomes with unchosen alternatives, enabling recall, revision, and reuse Chen & Wilson (2023). This intuition can be formalized by showing that residual invariants emerging from symmetry breaking necessarily take the form of closed cycles, which persist as homology classes and provide the structural foundation of memory.

**Lemma 1** (Symmetry Breaking Generates Invariant Cycles)**.** *Let a system evolve on a state space $\mathcal{Z}$ with symmetry group $G$. Suppose a perturbation $\varepsilon$ breaks $G$-equivariance by forcing the selection of a representative state $\Phi_\varepsilon$. Then: 1) The broken symmetry induces residual structures (orbits) invariant under residual transformations $H \subset G$. 2) These residual invariants manifest as closed cycles $\gamma \subset \mathcal{Z}$ stabilized by feedback (i.e. $\partial\gamma = 0$). 3) $\gamma$ defines a homology class $[\gamma] \in H_k(\mathcal{Z})$ that is stable under perturbations of $\varepsilon$, formalizing memory persistence.*

The proof for the above lemma can be found in Appendix A. This lemma establishes that symmetry breaking inevitably leaves behind residual invariants in the form of cycles, which act as stable memory traces of past selections. To fully understand their cognitive function, one must ask: What advantage does the system gain from organizing dynamics into such closed cycles? The key lies in the fact that cycles identify equivalence classes of trajectories, collapsing many superficially different paths into the same topological invariant Hatcher (2002). In other words, once dynamics are organized into homology classes, prediction and memory no longer depend on the precise *order* of steps, but only on the closure of the cycle due to the Abelian property of addition operators. This observation leads directly to the following theorem: cycles serve as the structural basis of *order invariance*, ensuring robustness in navigation, perception, action, and more abstract cognitive computations Hawkins (2021).

**Theorem 2** (Cycles Encode Order Invariance)**.** *Let $(\mathcal{Z}, x_0)$ be a pointed state space (latent manifold or graph) with base state $x_0$ ("home"). Let $\mathcal{A} = \{a_1, \ldots, a_m\}$ denote a finite set of local moves inducing paths $\{\alpha_i\}$ starting and ending in a neighborhood of their endpoints. For any finite sequence of moves $w = a_{i_1} \cdots a_{i_k}$ that yields a cycle $\gamma_w$ at $x_0$ (i.e., a homing trajectory), the first homology class $[\gamma_w] \in H_1(\mathcal{Z}; \mathbb{Z})$ depends only on the* multiset *of moves used (and their net orientations), not on their order. Equivalently, all order permutations of $w$ that remain cycles at $x_0$ determine the same element in $H_1$.*

Theorem 2 establishes that once trajectories are organized into cycles, their predictive value no longer depends on the precise ordering of steps but only on the closure of the cycle. This reduction

reflects a deeper topological dichotomy in memory formation. Algebraically, the identity $\partial^2 = 0$ ensures that boundaries of boundaries vanish Edelsbrunner & Harer (2010): incomplete chains cannot accumulate meaning unless they close, and only closed cycles can survive as invariants. Cognitively, this corresponds to the fact that exploratory fragments either collapse into trivial points (dots) with no relational content, or are stabilized into nontrivial cycles that encode order-invariant memory Babichev et al. (2025). In this sense, $\partial^2 = 0$ acts as the algebraic filter that separates forgotten scaffolds from consolidated invariants. To make this distinction explicit, we now formalize the roles of $H_0$ and $H_1$ in the following lemma (refer to Fig. 2).

**Dot–Cycle Dichotomy.** At the chain level, a "dot" (0–simplex) records isolated content, whereas a "cycle" (1–cycle) captures a closed relation in which endpoints cancel. The rule $\partial^2 = 0$ formalizes this passage: boundaries of fragments do not compose, but pairwise cancellation at endpoints yields a cycle that survives in homology. Cognitively, this is the move from token to trace Spens & Burgess (2024): contents $\Phi$ are registered as dots, yet only when linked by contextual relations $\Psi$ into a closed cycle do they consolidate as durable memory. Details regarding biological implementations can be found in Appendix B.

**Lemma 2** ($\partial^2 = 0$ Enforces the Dot-cycle Dichotomy). *Let $C_*(\mathcal{Z})$ denote the chain complex of a neural state space $\mathcal{Z}$. The homological identity $\partial^2 = 0$ implies that: 1) Any open chain $\sigma \in C_1(\mathcal{Z})$ with $\partial\sigma \neq 0$ must collapse to a trivial 0-cycle in $H_0(\mathcal{Z})$, encoding mere connectivity without relational content. 2) Any closed chain $\gamma \in C_1(\mathcal{Z})$ with $\partial\gamma = 0$ defines a homology class $[\gamma] \in H_1(\mathcal{Z})$. If $\gamma$ is not the boundary of a higher-dimensional chain, it represents a nontrivial cycle that persists as a stable memory trace. Thus, $\partial^2 = 0$ acts as a topological filter: boundaries of boundaries vanish, ensuring that only two outcomes are possible, collapse into trivial dots ($H_0$) or persistence as nontrivial cycles ($H_1$).*

Lemma 2 provides the *algebraic gate* for memory: $\partial^2 = 0$ prunes open, order-sensitive fragments and admits only closed loops as meaningful carriers. To connect this structural pruning with predictive power, we now view closure through an information-theoretic lens Cover (1999). When many orderings of the same events are possible, their variability behaves as symmetry-induced noise. Closure collapses these degrees of freedom onto a residual loop, thereby concentrating probability mass on what is repeatable and compressing description length. In effect, cycles are the *sufficient statistics* of paths: once a trajectory closes, order fluctuations become irrelevant for forecasting Friston (2018). The algebraic identity $\partial^2 = 0$ has an information-theoretic counterpart: broken symmetry reduces entropy by collapsing many equivalent paths into one invariant cycle. The next proposition formalizes this entropy–prediction link via symmetry breaking that leaves an invariant cycle.

**Proposition 1** (Entropy Minimization Improves Prediction by Cycles). *Let a system generate trajectories in a state space $\mathcal{Z}$. Suppose initially, the system has a symmetry $G$ (e.g. different orders of moves or observations are treated as equivalent). A perturbation breaks this full symmetry, but leaves behind an invariant cycle $\gamma \subset \mathcal{Z}$ with $\partial\gamma = 0$. Then we have: 1) The cycle $\gamma$ encodes what is stable across different orders or paths; 2) Predictions about future outcomes need only depend on $\gamma$ (and context), not on the detailed order of past steps; 3) Thus, broken symmetry reduces noise from order-specific variations and improves prediction by preserving only what remains invariant.*

Proposition 1 identifies *what* survives order variability: the residual invariant cycle $\gamma$. To pass from structure to statistics, note that discarding order-specific fluctuations is equivalent to an entropy drop: probability mass that was spread over many orderings is reassigned to the closed loop that summarizes them. In a non-ergodic system, this manifests as *measure concentration* on the surviving cycles Ledoux (2001). Therefore, predictive sufficiency (dependence only on $[\gamma]$) coincides with entropy reduction (symmetry breaking) and with the asymptotic concentration of $\mu_t$ on invariant classes. The following corollary makes this equivalence explicit.

**Corollary 1** (Prediction as Concentration on Cycles). *For a non-ergodic system $(X, T)$, prediction is possible iff the probability measure $\mu_t$ concentrates on invariant cycles $[\gamma] \in H_k(X)$ as $t \to \infty$. Equivalently, **Prediction** $\iff$ **Entropy Reduction via Symmetry Breaking** $\iff$ **Measure Concentration on Cycles**. Therefore, the structural invariants revealed by broken symmetry are precisely the carriers of predictive information, ensuring reliable memory and generalization across time.*

Corollary 1 identifies *what* supports prediction: global dynamics must collapse onto persistent cycles. *How* such cycles arise is local: symmetry breaking forces a choice among equivalent alterna-

tives, and the discarded possibilities are reorganized into recurrent loops. These loops stabilize the selected outcome while retaining counterfactual access, thereby creating the invariant structures that concentrate probability mass and convert uncertainty into predictive stability.

# 4 MEMORY-AMORTIZED INFERENCE FOR TOPOLOGICAL CLOSURE

To operationalize this picture in cognition, we adopt the *Context–Content Uncertainty Principle (CCUP)* Li (2025a): stable memory traces correspond to low-entropy *content variables* $\Phi$ (persistent homological cycles), while transient variability is captured by high-entropy *context variables* $\Psi$. In what follows, we show how *Memory–Amortized Inference (MAI)* implements cycle formation by holding $\Phi$ fixed as reusable structure and adapting $\Psi$ until residual boundaries cancel ($\partial^2 = 0$), thereby achieving topological closure.

**Content variable $\Phi$ as low-entropy homology.** Within CCUP, the content variable $\Phi$ corresponds to information that is both specific and stable. Mathematically, $\Phi$ is identified with nontrivial homology classes: cycles $[\gamma] \in H_k(\mathcal{Z})$ that cannot be reduced to boundaries. Such cycles encode persistent, low-entropy structures because many possible trajectories or micro-states collapse into the same equivalence class. In neural terms, $\Phi$ reflects patterns of activity that recur reliably across different contexts, such as a learned motor primitive, a familiar spatial route, or a well-established object representation. By filtering away order-dependent variability, $\Phi$ preserves only the invariant relational structure that remains after symmetry breaking. This makes $\Phi$ the stable substrate of memory and the carrier of predictive power: once identified, it can be recalled, reused, and composed into higher-order cognitive structures.

**Context variable $\Psi$ as high-entropy scaffolding.** In contrast, the context variable $\Psi$ captures the transient, exploratory, and often noisy aspects of cognition. Topologically, $\Psi$ is associated with trivial cycles or short-lived features in the persistence barcode: loops that quickly vanish under perturbation or deformation. These cycles act as *scaffolding*, supporting the discovery and stabilization of $\Phi$ but not themselves persisting as memory. In information-theoretic terms, $\Psi$ is high-entropy: it reflects a large space of possibilities, many of which will be pruned away as the system concentrates its measure on low-entropy $\Phi$ structures. Biologically, $\Psi$ is implemented by slow, contextual rhythms (e.g. theta oscillations) or exploratory neural activity that supplies diverse scaffolds for binding. Through dynamic alignment and phase-resetting, these high-entropy contextual structures are folded into persistent content loops, allowing cognition to maintain flexibility while ensuring stability in memory formation.

Taken together, $\Phi$ and $\Psi$ form a complementary pair: $\Phi$ supplies the order-invariant backbone that can be reused across contexts, while $\Psi$ provides the exploratory variability from which such backbones are discovered. CCUP therefore prescribes an operational loop: hold candidate content steady, let context range, and accept only those pairings that close into cycles (i.e., cancel boundaries). This suggests a general law of cognitive economy in which *structure leads* and *specificity follows*: stable invariants guide, while transient scaffolds adapt until closure is achieved. We now make this heuristic precise as a principled statement.

**Principle 3** (Structure-Before-Specificity Principle). *Let $\Phi$ denote low-entropy content variables corresponding to nontrivial homology classes $[\gamma] \in H_k(\mathcal{Z})$, and let $\Psi$ denote high-entropy contextual scaffolds corresponding to transient or trivial cycles. Then cognition obeys the following principle: 1) (**Structure before specificity**) Stable content $\Phi$ arises from nontrivial cycles that persist across perturbations. These cycles define the backbone of memory and predictive power. 2) (**Specificity from scaffolding**) Context $\Psi$ supplies a high-entropy exploratory substrate: transient cycles that may collapse but provide the variability needed to refine, adapt, or recombine $\Phi$. 3) (**Dynamic alignment**) The interaction of $\Psi$ and $\Phi$ via cycle closure ($\partial^2 = 0$) ensures that contextual exploration is funneled into persistent content loops, transforming noisy scaffolds into stable memory traces.*

The above principle prescribes an operational recipe: stabilize $\Phi$ as reusable structure and let $\Psi$ explore until closure cancels residual boundaries. *Memory–amortized inference (MAI)* is the algorithmic embodiment of this recipe. Instead of re-solving each inference problem from scratch, MAI retrieves a candidate invariant (a cycle-level template for $\Phi$), then adapts $\Psi$ until the pair $(\Psi, \Phi)$

closes (i.e., $\partial^2 = 0$), pruning order-specific noise. In effect, $\Phi$ functions as a low-entropy prior over solutions, while $\Psi$ supplies the high-entropy search that is guided and terminated by topological closure. We formalize MAI as a general strategy for reducing the computational cost of inference by storing and reusing structured latent representations. The key idea is to construct a memory of prior inference results such that new inference problems can be approximated by querying and adapting from this memory, rather than solving the full problem from scratch. Let $\Psi \in \mathcal{X}$ denote the observable context and $\Phi \in \mathcal{S}$ the latent content to be inferred. Let $\mathcal{L}(\Psi, \Phi)$ denote a loss or cost function encoding the fidelity or predictive value of $\Phi$ under context $\Psi$. We assume that inference corresponds to solving the following optimization: $\Phi^* = \arg\min_{\Phi \in \mathcal{S}} [\mathcal{L}(\Psi, \Phi)]$. Formally, we start with the following definition (refer to Fig. 3).

**Definition 2** (Memory-Amortized Inference). *Let $\mathcal{M} = \{(\Psi^{(i)}, \Phi^{(i)})\}_{i=1}^N$ be a memory of prior context–content pairs, and let $\mathcal{R} : \mathcal{X} \times \mathcal{M} \to \mathcal{S}$ be a retrieval-and-adaptation operator and $\mathcal{F} : \mathcal{S} \times \mathcal{X} \to \mathcal{S}$ be the bootstrapping update operator implemented via generative simulation. Inference is said to be* memory-amortized *if it is formulated as a structural cycle between* content $\Phi$ *and* context $\Psi$*, where memory acts as a reusable substrate for inference: $\Phi_{t+1} = \mathcal{F}(\Phi_t, \Psi_t)$, $\Phi_t \approx \mathcal{R}(\Phi_{t+1}, \Psi_t)$ in lieu of directly optimizing $\Phi^*$, such that the expected cost satisfies $\mathbb{E}_\Psi \left[ \mathcal{L}(\Psi, \hat{\Phi}) \right] \leq \mathbb{E}_\Psi [\mathcal{L}(\Psi, \Phi^*)] + \varepsilon$, for some amortization gap $\varepsilon \ll \mathcal{L}(\Psi, \cdot)$, and where the runtime cost of $\mathcal{R}$ is substantially lower than full inference.*

**Memory-Amortized Inference Cycle**

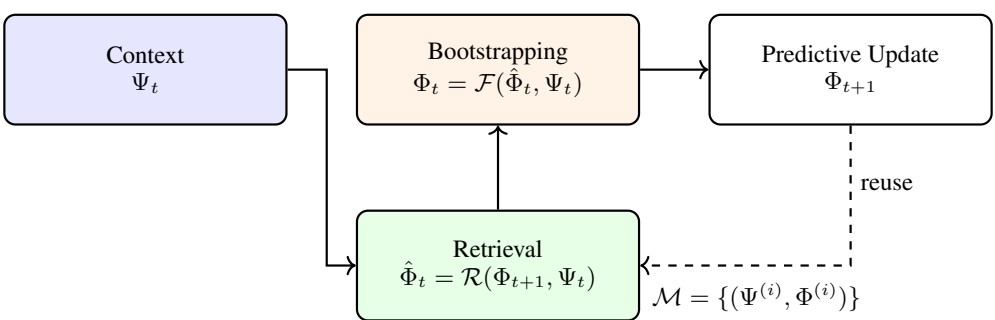

Figure 3: Cycle of MAI. Instead of recomputing $\Phi^* = \arg\min \mathcal{L}(\Psi, \Phi)$, the system reuses prior trajectories: $\Phi_{t+1}$ and $\Psi_t$ guide memory-based retrieval via $\mathcal{R}$, and bootstrapping $\mathcal{F}$ updates the latent state $\Phi_t$. The process forms a self-consistent loop grounded in structured memory.

**The Retrieval-and-Adaptation Operator $\mathcal{R}$.** The retrieval-and-adaptation operator $\mathcal{R} : \mathcal{X} \times \mathcal{M} \to \mathcal{S}$ serves as the core mechanism by which inference avoids re-computation. Given an input query (typically latent or perceptual), $\mathcal{R}$ retrieves relevant elements from the memory $\mathcal{M} = \{(\Psi^{(i)}, \Phi^{(i)})\}_{i=1}^N$ and performs a lightweight adaptation to generate a candidate solution $\hat{\Phi}$. Operationally, $\mathcal{R}$ consists of two stages: 1) **Retrieval:** Identify a relevant subset of memory entries $\{(\Psi^{(j)}, \Phi^{(j)})\} \subset \mathcal{M}$ based on similarity to the current context $\Psi_t$. This can be performed via kernel-based attention, similarity search in latent space, or topological proximity under homological constraints. 2) **Adaptation:** Modulate or interpolate the retrieved $\Phi^{(j)}$ values conditioned on $\Psi_t$, resulting in a candidate $\hat{\Phi}_t = \mathcal{R}(\Phi_{t+1}, \Psi_t)$. This step often involves gradient-free adjustments (e.g., feature warping, parameter blending) and is significantly cheaper than full inference.

The *retrieval-and-adaptation operator $\mathcal{R}$* in MAI generalizes the classical notion of key-value memory used in neural attention and memory-augmented models. In conventional key-value memory systems Weston et al. (2014); Sukhbaatar et al. (2015), memory is structured as a set of key-value pairs: $\mathcal{M} = \{(\Psi^{(i)}, \Phi^{(i)})\}_{i=1}^N$, where a context vector $\Psi$ acts as a *key* to retrieve values $\Phi$ via similarity-based soft addressing: $\hat{\Phi} = \sum_i w_i \Phi^{(i)}$, $w_i = \frac{\exp(-d(\Psi, \Psi^{(i)}))}{\sum_j \exp(-d(\Psi, \Psi^{(j)}))}$. This model supports one-shot retrieval but lacks structural consistency or bidirectional inference. By contrast, the operator $\mathcal{R}(\Phi_{t+1}, \Psi_t; \mathcal{M})$ in MAI performs a more general operation: it *retrieves* a candidate latent representation from memory based on both the current context $\Psi_t$ and a target latent code $\Phi_{t+1}$, and then *adapts* it to produce a consistent approximation of the preceding latent

state $\Phi_t$. This supports inference in reverse time and satisfies the memory-amortized constraint: $\Phi_t \approx \mathcal{R}(\Phi_{t+1}, \Psi_t), \quad \Phi_{t+1} = \mathcal{F}(\Phi_t, \Psi_t)$. The operator $\mathcal{R}$ thereby enables cycle-consistent inference, crucial for temporal coherence and structural reuse. Unlike key-value memory, which operates over flat vector spaces, $\mathcal{R}$ may act over structured memory (e.g., graphs, latent manifolds, or topological complexes) and is inherently adaptive. A summary of the distinction is provided below:

**The Bootstrapping Update Operator $\mathcal{F}$.** The bootstrapping operator $\mathcal{F} : \mathcal{S} \times \mathcal{C} \to \mathcal{S}$ governs the internal dynamics of inference by iteratively updating the latent content representation $\Phi_t$ given the context $\Psi_t$. It defines a recurrence: $\Phi_{t+1} = \mathcal{F}(\Psi_t, \Psi_t)$, where $\mathcal{F}$ encodes the system's structural prior, capturing the directionality, topology, and dynamic consistency of inference over time. Unlike standard update rules that minimize a loss from scratch, $\mathcal{F}$ performs bootstrapping: each update is initialized from a prior memory-induced state, often already close to the optimal solution due to cycle recurrence. Here are several key properties of $\mathcal{F}$: 1) **Cycle-Consistency:** If $(\Phi_t, \Psi_t) \in \gamma$ for some memory cycle $\gamma \subset \mathcal{Z}$, then $\Phi_{t+T} \approx \Phi_t$, enabling amortization via structural recurrence. 2) **Structural Biasing:** Updates follow latent paths constrained by prior topology (e.g., flow fields over homology classes or attention-modulated latent graphs), enforcing low-entropy generalization. 3) **Minimal Cost Gradient**: Because the initialization $\Phi_t$ already lies near an attractor, the subsequent update $\Phi_{t+1}$ requires only a small corrective shift, further amortizing the inference process.

The bootstrapping update operator $\mathcal{F}$ in MAI is structurally analogous to the *half-step down* trick used in Q-learning Watkins & Dayan (1992) and temporal difference (TD) methods Sutton & Barto (1998). In Q-learning, the value function is updated by approximating the current value via a one-step lookahead: $Q(s_t, a_t) \leftarrow r_t + \gamma \max_{a'} Q(s_{t+1}, a')$, which yields the approximation $Q(s_t) \approx Q(s_{t+1})$. This forward-directed value propagation allows reinforcement learning agents to estimate long-term outcomes without simulating entire trajectories. By contrast, MAI reverses the time direction: the update operator $\mathcal{F}$ bootstraps latent inference forward using structured memory and contextual cues: $\Phi_{t+1} = \mathcal{F}(\Phi_t, \Psi_t)$, and this is inverted by retrieval: $\Phi_t \approx \mathcal{R}(\Phi_{t+1}, \Psi_t)$. This dual relationship forms the backbone of the MAI half-step trick: the current latent content $\Phi_t$ generates the next-step prediction $\Phi_{t+1}$, which in turn can be used to reconstruct $\Phi_t$. While Q-learning bootstraps value via reward-driven transitions, MAI bootstraps inference through latent memory and context, yielding a cycle-consistent structure that reduces entropy. Both approaches use bootstrapping to manage uncertainty and amortize computational cost, but in opposite directions, highlighting a deeper time-reversed duality between learning and inference (refer to Appendix C). This recursive formulation enables stable inference trajectories that converge toward contextually relevant attractors, effectively amortizing the cost of learning across time. The underlying dynamics of this process can be formalized as a contractive map over a structured retrieval cycle, leading to provable convergence under mild assumptions. We now state the following result, which captures the fixed-point stability of the MAI loop:

**Proposition 2** (Topological Closure via Structural Recursion). *Let $\mathcal{T}(\Phi, \Psi) := \mathcal{F}(\mathcal{R}(\Phi, \Psi), \Psi)$ be the composite update in MAI. Suppose $\mathcal{T}$ is contractive in its first argument for fixed context $\Psi$. Then there exists a unique fixed point $\Phi^*$ such that: $\Phi^* = \mathcal{T}(\Phi^*, \Psi)$ Moreover, the inference trajectory $\{\Phi_t\}_{t=0}^{\infty}$ forms a closed loop in latent space as: $\lim_{t \to \infty} \|\Phi_t - \Phi^*\| = 0$ This latent recurrence corresponds to a nontrivial 1-cycle, representing topological closure in the MAI manifold.*

Proposition 2 establishes closure at the level of latent dynamics: a contractive structural recursion yields a fixed point and a recurrent trajectory that "homes" to it, i.e., a geometric 1-cycle in the MAI manifold. We now lift this geometric closure to the algebraic level of chains. Specifically, the same retrieve–update loop can be read as a chain-homotopy correction that cancels residual boundaries in the context–content complex. In this view, latent recurrence (fixed-point closure) and homological recurrence (boundary cancellation) are two faces of the same mechanism. The next theorem formalizes this equivalence by showing that MAI implements topological closure via $\partial^2 = 0$ (its proof can be found in Appendix A).

**Theorem 3** (MAI as Computational Realization of Topological Closure). *Let $(C_\bullet, \partial)$ be a chain complex encoding context–content relations, with $\Psi$ as high-entropy scaffolds and $\Phi$ as candidate content variables. In Memory-Amortized Inference (Definition 1), the iterative cycle $\Phi_{t+1} = \mathcal{F}(\Phi_t, \Psi_t), \quad \Phi_t \approx \mathcal{R}(\Phi_{t+1}, \Psi_t)$ implements a homotopy update that cancels residual boundaries: $\partial(\Psi_t, \Phi_t) \mapsto \partial(\Psi_{t+1}, \Phi_{t+1}) \approx 0$. Thus, amortization prunes misaligned, order-dependent fragments (open boundaries) and preserves only reproducible cycles $[\gamma] \in H_k(C_\bullet)$. Equivalently, MAI realizes topological closure by enforcing $\partial^2 = 0$ in computation: context–content updates that fail to close are discarded, while those that re-enter memory persist as invariants.*

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

## A   Proofs of Lemmas, Theorems and Propositions

**Proof of Lemma 1**

*Proof (sketch).* Let $\mathcal{Z}$ be a smooth manifold with a smooth (left) action of a Lie group $G$, and let $f_\varepsilon : \mathcal{Z} \to T\mathcal{Z}$ be a $C^1$ family of vector fields such that $f_0$ is $G$–equivariant ($f_0(g \cdot z) = Dg \cdot f_0(z)$) and $\varepsilon \mapsto f_\varepsilon$ breaks equivariance to a proper subgroup $H \subset G$ by selecting a representative $\Phi_\varepsilon$ in each $G$–orbit near $\Phi_0$.

*(1) Residual invariant structure.* At $\varepsilon = 0$, the $G$–equivariance implies that the $G$–orbit $\mathcal{O}_0 := G \cdot \Phi_0$ is invariant for $f_0$. For $|\varepsilon|$ small, equivariance with respect to $H$ persists, and the *residual $H$–orbit* $\mathcal{O}_\varepsilon := H \cdot \Phi_\varepsilon \subset \mathcal{Z}$ is $f_\varepsilon$–invariant. By the slice theorem, a neighborhood of $\mathcal{O}_0$ is $G$–equivariantly diffeomorphic to $G \times_H S$ for some slice $S$, hence the residual structure is modeled on the homogeneous space $G/H$ near $\Phi_\varepsilon$.

*(2) Emergence of closed cycles under feedback.* Assume a stabilizing feedback (or dissipation) renders $\mathcal{O}_\varepsilon$ *normally hyperbolic*. Then the invariant manifold $\mathcal{O}_\varepsilon$ persists for small $\varepsilon$ and the restricted flow $f_\varepsilon|_{\mathcal{O}_\varepsilon}$ is $H$–invariant. If $\pi_1(\mathcal{O}_\varepsilon) \neq 0$ (e.g. when $\mathcal{O}_\varepsilon$ contains an $S^1$ factor, as is typical for residual phase symmetries), there exist periodic orbits $\gamma \subset \mathcal{O}_\varepsilon \subset \mathcal{Z}$ representing nontrivial classes in $\pi_1$. As 1–chains, periodic orbits are cycles, hence $\partial\gamma = 0$ in the singular chain complex. More generally, if the residual invariant manifold contains an embedded $k$–dimensional compact submanifold $N^k \subset \mathcal{O}_\varepsilon$ invariant under the restricted flow, its fundamental class yields a closed $k$–cycle in $C_k(\mathcal{Z})$.

*(3) Homological persistence under perturbation.* Normal hyperbolicity plus smooth dependence on parameters implies that $\mathcal{O}_\varepsilon$ and its periodic orbits (or invariant submanifolds) vary continuously for small $\varepsilon$; hyperbolic periodic orbits persist (Structural Stability). Consequently, any closed chain $c_\varepsilon$ carried by $\mathcal{O}_\varepsilon$ depends continuously on $\varepsilon$ and its homology class $[c_\varepsilon] \in H_k(\mathcal{Z})$ is invariant under the induced homotopy. Hence $[\gamma] \in H_k(\mathcal{Z})$ is stable for small perturbations, formalizing memory persistence.

Combining (1)–(3) proves the three claims: symmetry breaking selects a residual invariant structure, feedback stabilizes closed cycles on it ($\partial\gamma = 0$), and the resulting homology classes persist under small perturbations of $\varepsilon$. $\square$

**Proof of Lemma 2**

*Proof.* Write $Z_k := \ker(\partial_k)$ and $B_k := \operatorname{im}(\partial_{k+1})$ so that $H_k(\mathcal{Z}) = Z_k/B_k$ and $\partial_{k-1} \circ \partial_k = 0$ for all $k$.

(1) Let $\sigma \in C_1(\mathcal{Z})$ with $\partial\sigma \neq 0$. Then $\sigma \notin Z_1$, so it cannot define a class in $H_1$. The only homological information it induces is via its boundary $\partial\sigma \in C_0(\mathcal{Z})$. But by definition $\partial\sigma \in B_0 = \operatorname{im}(\partial_1)$, and since $H_0 = Z_0/B_0$ with $B_0 \subseteq Z_0$ (because $\partial^2 = 0$), we have $[\partial\sigma] = 0$ in $H_0$. Thus the open chain contributes no nontrivial $H_1$ content and collapses, at best, to the trivial $H_0$ class that encodes mere connectivity (membership in a component), not a relational invariant.

(2) Let $\gamma \in C_1(\mathcal{Z})$ with $\partial\gamma = 0$. Then $\gamma \in Z_1$ and its homology class $[\gamma] \in H_1 = Z_1/B_1$ is well-defined. If moreover $\gamma \notin B_1 = \operatorname{im}(\partial_2)$, then $[\gamma] \neq 0$ in $H_1$, i.e., $\gamma$ represents a nontrivial 1-cycle. Such a class is invariant under addition of boundaries ($\gamma \sim \gamma + \partial c_2$), hence persists under deformations that do not cross a filling 2-chain, formalizing stability of the memory trace.

Finally, $\partial^2 = 0$ implies $B_k \subseteq Z_k$ for all $k$, so every boundary is a cycle but not conversely. Consequently any 1-chain is either (i) non-closed, in which case it reduces to a trivial element in $H_0$, or (ii) closed, in which case it defines a class in $H_1$ that is nontrivial precisely when it is not a boundary. This is the dot–cycle dichotomy. $\square$

**Proof of Theorem 1**

*Proof.* We give a standard proof in simplicial (or singular) homology, then note two equivalent formulations (cubical/differential forms) for completeness.

**Simplicial chains.** Let $C_k$ be the free abelian group generated by oriented $k$-simplices $\sigma = [v_0, \ldots, v_k]$ of an oriented simplicial complex. Define the boundary operator $\partial_k : C_k \to C_{k-1}$ by

$$\partial_k[v_0, \ldots, v_k] = \sum_{i=0}^{k}(-1)^i [v_0, \ldots, \widehat{v_i}, \ldots, v_k],$$

where the hat indicates omission and each face inherits the induced orientation. Apply $\partial_{k-1}$ once more:

$$\partial_{k-1}\partial_k[v_0, \ldots, v_k] = \sum_{i=0}^{k}(-1)^i \sum_{j=0}^{k-1}(-1)^j [v_0, \ldots, \widehat{v_i}, \ldots, \widehat{v_{i+j'}}, \ldots, v_k],$$

where $j'$ denotes the corresponding original index in $\{0, \ldots, k\} \setminus \{i\}$.

Every $(k-2)$-face of $[v_0, \ldots, v_k]$ arises *twice* in this double sum: once by deleting $v_i$ then $v_j$ with $i < j$, and once by deleting $v_j$ then $v_i$. These two occurrences have opposite signs and thus cancel. Formally, fix $0 \leq i < j \leq k$. The face obtained by deleting $v_i$ then $v_j$ appears with sign $(-1)^i(-1)^{j-1} = (-1)^{i+j-1}$, while deleting $v_j$ then $v_i$ yields sign $(-1)^j(-1)^i = (-1)^{i+j}$. Hence the two contributions sum to zero:

$$(-1)^{i+j-1} + (-1)^{i+j} = 0.$$

Since every $(k-2)$-face of $\sigma$ appears exactly in such canceling pairs, all terms vanish and therefore $\partial_{k-1}\partial_k = 0$, i.e. $\partial^2 = 0$.

**Singular chains (same combinatorics).** For singular homology, $C_k$ is generated by singular simplices $\sigma : \Delta^k \to X$, and the boundary uses face inclusions $d_i : \Delta^{k-1} \hookrightarrow \Delta^k$:

$$\partial_k\sigma = \sum_{i=0}^{k}(-1)^i \sigma \circ d_i.$$

Then

$$\partial_{k-1}\partial_k\sigma = \sum_{i<j} \left((-1)^{i+j} \sigma \circ d_i \circ d_{j-1} + (-1)^{i+j} \sigma \circ d_j \circ d_i\right) = 0,$$

because $d_i \circ d_j = d_{j+1} \circ d_i$ and the two terms cancel in pairs.

**Cubical chains (face maps).** In cubical homology, the boundary is an alternating sum of front/back faces along each coordinate. The same "each $(k-2)$-face appears twice with opposite sign" cancellation proves $\partial^2 = 0$.

**Differential forms (Stokes $\Rightarrow d^2 = 0$).** On smooth manifolds, Stokes' theorem implies $\int_{\partial(\partial\Omega)} \omega = \int_\Omega d(d\omega)$. Since $\partial \circ \partial = 0$ as a current, it follows that $d^2 = 0$. By de Rham's theorem this is dual to the chain-level statement $\partial^2 = 0$.

**Cognitive interpretation (corollary).** Because $\partial^2 = 0$, any attempt to "promote" boundary fragments (order- and context-dependent specifics) to stable carriers necessarily eliminates dangling inconsistencies: open edges are paired and cancel, and only *closed* cycles persist. These persistent cycles are precisely the invariants that can be stored as memory and reused for prediction. Hence the closure identity guarantees the existence of stable semantic carriers and underwrites the claim that intelligence (as memory-based prediction) rests on cycle closure.

This completes the proof that the boundary of a boundary vanishes: $\partial^2 = 0$. □

**Proof of Theorem 2**

*Proof:* The key insight is the Abelian property of the addition operator. Concatenate local moves to form cycles based at $x_0$, producing elements of the fundamental group $\pi_1(\mathcal{Z}, x_0)$. The Hurewicz map $h : \pi_1(\mathcal{Z}, x_0) \to H_1(\mathcal{Z}; \mathbb{Z})$ abelianizes path composition: commutators vanish in $H_1$. Hence for cycles $\gamma, \eta$, $[\gamma \cdot \eta] = [\eta \cdot \gamma]$ and, more generally, any permutation of cycle segments yields the same homology class, provided the path remains closed. Thus $[\gamma_w]$ is invariant to the *order* of constituent

moves and depends only on their cumulative 1-chain (the signed sum of traversed edges/segments). Intuitively, homology collapses all order-specific reparameterizations and commutator structure, retaining only the closed-cycle content. □

**Proof of Theorem 3**

*Proof.* Let $(C_\bullet, \partial)$ encode context–content relations and write the *residual boundary* at step $t$ as $r_t := \partial(\Psi_t, \Phi_t) \in C_{k-1}$. Define the MAI update $\mathcal{U} := \mathcal{R} \circ \mathcal{F}$ so that $(\Psi_{t+1}, \Phi_{t+1}) = \mathcal{U}(\Psi_t, \Phi_t)$ with $\Phi_{t+1} = \mathcal{F}(\Phi_t, \Psi_t)$ and $\Phi_t \approx \mathcal{R}(\Phi_{t+1}, \Psi_t)$. Assume (i) *boundary-aware* updates: there exists a linear operator $H : C_k \to C_{k+1}$ (a homotopy) and $\eta \in (0, 1]$ such that, up to the amortization error $\epsilon_t$,

$$(\Psi_{t+1}, \Phi_{t+1}) = (\Psi_t, \Phi_t) - \eta\, H\, r_t + \epsilon_t, \qquad \|\epsilon_t\| \leq \varepsilon,$$

and (ii) $\partial^2 = 0$ on $C_\bullet$. Then

$$r_{t+1} = \partial(\Psi_{t+1}, \Phi_{t+1}) = \partial(\Psi_t, \Phi_t) - \eta\, \partial H\, r_t + \partial \epsilon_t = (I - \eta\, \partial H) r_t + \partial \epsilon_t.$$

Choose $H$ so that $P := I - \partial H - H \partial$ is the standard chain-homotopy projector onto $Z_k := \ker \partial$ (e.g. a Moore–Penrose choice on a chosen splitting). Using $\partial^2 = 0$,

$$r_{t+1} = (I - \eta\, \partial H) r_t + \partial \epsilon_t = (P + H\partial - \eta\, \partial H) r_t + \partial \epsilon_t = (I - \eta)\, r_t + \partial \epsilon_t,$$

since $P r_t = 0$ and $\partial r_t = \partial^2(\cdot) = 0$. Hence $\|r_{t+1}\| \leq (1 - \eta) \|r_t\| + \|\partial\|\, \varepsilon$. If $\eta \in (0, 1]$ and $\varepsilon$ is the small amortization gap from Definition 1, the residuals converge: $\|r_t\| \to 0$ as $t \to \infty$ (exactly if $\varepsilon = 0$, or to an $O(\varepsilon)$ neighborhood otherwise). Thus, any limit point $(\Psi_\infty, \Phi_\infty)$ satisfies $\partial(\Psi_\infty, \Phi_\infty) = 0$, i.e. it lies in $Z_k$ and represents a closed cycle $[\gamma] \in H_k$. Moreover, because $\mathcal{R}$ retrieves from memory and $\mathcal{F}$ bootstraps by simulation while satisfying the amortization inequality $\mathbb{E}_\Psi[\mathcal{L}(\Psi, \hat{\Phi})] \leq \mathbb{E}_\Psi[\mathcal{L}(\Psi, \Phi^*)] + \varepsilon$, open, order-dependent trajectories (with large $\|r_t\|$) are not retained, while reproducible closures are. Therefore MAI acts as a homotopy-based projection onto $\ker \partial$, canceling boundaries and preserving precisely the invariant cycles, i.e. it realizes topological closure computationally. □

**Proof of Proposition 1**

*Proof (sketch).* Let $\mathcal{P}$ be the set of finite trajectories (paths) in $\mathcal{Z}$ and let $G$ act on $\mathcal{P}$ by the symmetry that permutes orderings of local moves/observations. Define an equivalence relation $p \sim_G p'$ iff $p' = g \cdot p$ for some $g \in G$. Suppose a perturbation breaks $G$ to a residual subgroup $H$ and induces a continuous, $H$–invariant map

$$q : \mathcal{P} \longrightarrow Z_k(\mathcal{Z}) / B_k(\mathcal{Z}) \cong H_k(\mathcal{Z}), \quad p \mapsto [\gamma(p)],$$

that sends each path $p$ to the homology class of its closing cycle $\gamma(p)$ (if $p$ does not close, $q$ maps it to the trivial class). By assumption there exists a nontrivial invariant cycle $\gamma$ with $\partial \gamma = 0$ that survives the perturbation (i.e. $[\gamma] \neq 0$ and $H$–invariant).

(1) *Cycle encodes order-invariant stability.* If $p' \sim_G p$, then $q(p') = q(p)$ because permutations of the same local moves that remain closable yield homologous loops. Hence $[\gamma]$ is constant on $G$–orbits and captures precisely what is invariant under order rearrangements. This proves (1).

(2) *Predictive sufficiency of $[\gamma]$.* Let $Y$ denote a future outcome (or next observation) to be predicted from the past path $P \in \mathcal{P}$ and any ambient context variable $C$ (slow parameters). Assume the perturbation enforces the residual symmetry so that *conditional on the homology class* we have

$$\mathbb{P}(Y \mid P, C) = \mathbb{P}(Y \mid q(P), C).$$

That is, order-specific information in $P$ beyond its cycle class does not affect the conditional law of $Y$. Then $q(P)$ is a (Blackwell) sufficient statistic for predicting $Y$ given $C$. By the data-processing inequality,

$$I(Y; P \mid C) \geq I(Y; q(P) \mid C),$$

with equality under the displayed conditional independence, which shows that prediction needs only depend on the invariant $[\gamma] = q(P)$ (and $C$), not on the detailed order in $P$. This proves (2).

(3) *Entropy reduction improves prediction.* Let $\mathcal{G}$ be the $\sigma$-algebra generated by $P$ and let $\mathcal{G}_{\text{inv}}$ be that generated by $q(P)$. Since $q$ maps many orderings to the same class, $H(q(P)) \leq H(P)$ and $\mathbb{E}[\ell(Y, \hat{Y})]$ for any Bayes-optimal predictor $\hat{Y}$ is the same whether conditioning on $P$ or $q(P)$ (by sufficiency). Thus collapsing order-specific variability to $[\gamma]$ strictly reduces the description length of the predictor while preserving optimal predictive risk. Interpreting entropy as uncertainty, the symmetry breaking acts to concentrate probability mass onto $[\gamma]$–classes (measure concentration on invariant cycles), thereby removing order noise and improving generalization: predictions depend only on what remains invariant. This proves (3).

Altogether, the perturbation-induced residual symmetry yields an invariant cycle $\gamma$ (*closure $\partial\gamma = 0$*) whose class $[\gamma]$ summarizes all order permutations of closable paths; $[\gamma]$ is a sufficient statistic for forecasting $Y$ (given context), and the associated entropy drop reflects the elimination of order-specific noise. Hence entropy minimization via symmetry breaking improves prediction by preserving only invariant cycles. $\square$

**Proof of Proposition 2**

*Proof (sketch).* Fix a context $\Psi$ and define the self-map $\mathcal{T}_\Psi(\Phi) := \mathcal{F}(\mathcal{R}(\Phi, \Psi), \Psi)$. By assumption, $\mathcal{T}_\Psi$ is a contraction in its first argument on a complete metric space $(\mathcal{S}, \|\cdot\|)$: there exists $0 < \kappa < 1$ such that $\|\mathcal{T}_\Psi(\Phi) - \mathcal{T}_\Psi(\Phi')\| \leq \kappa \|\Phi - \Phi'\|$ for all $\Phi, \Phi' \in \mathcal{S}$.

*(Existence, uniqueness, and convergence).* By the Banach fixed-point theorem, there exists a unique fixed point $\Phi^* \in \mathcal{S}$ with $\Phi^* = \mathcal{T}_\Psi(\Phi^*)$, and for any initialization $\Phi_0$ the iterates $\Phi_{t+1} = \mathcal{T}_\Psi(\Phi_t)$ satisfy $\|\Phi_t - \Phi^*\| \leq \kappa^t \|\Phi_0 - \Phi^*\| \to 0$ as $t \to \infty$. This proves the first two claims.

*(Latent recurrence as a closed 1-cycle).* Form the polygonal 1-chain

$$c_n := \sum_{t=0}^{n-1} e_t \quad \text{with} \quad e_t := [\Phi_t, \Phi_{t+1}],$$

where $[\Phi_t, \Phi_{t+1}]$ denotes the oriented edge in latent space joining successive iterates. Its boundary is $\partial c_n = \Phi_n - \Phi_0$. Close the polygon by adding the short edges $e_n = [\Phi_n, \Phi^*]$ and $e_{-1} = [\Phi^*, \Phi_0]$ to obtain

$$\tilde{c}_n := c_n + e_n + e_{-1}, \qquad \partial\tilde{c}_n = (\Phi_n - \Phi_0) + (\Phi^* - \Phi_n) + (\Phi_0 - \Phi^*) = 0.$$

Thus each $\tilde{c}_n$ is a 1-cycle. Since $\|\Phi_n - \Phi^*\| \to 0$, the closing edges $e_n, e_{-1}$ have lengths $\to 0$, and the sequence $\{\tilde{c}_n\}$ converges (in the 1-chain norm induced by edge length) to a limit 1-chain $\gamma$ with $\partial\gamma = 0$. Hence the MAI trajectory defines a closed loop (a 1-cycle) in latent space.

*(Nontriviality and topological closure).* If the image of the trajectory lies in a region whose chosen 2-chain complex (e.g. a Vietoris–Rips or Čech complex at some scale $\varepsilon$) contains no filling 2-chain for $\gamma$, then $\gamma \notin \operatorname{im} \partial_2$ and $[\gamma] \neq 0$ in $H_1$, yielding a nontrivial cycle. This expresses *topological closure*: the structural recursion contracts to a fixed point while the induced 1-chain closes with vanishing boundary; nontriviality holds precisely when the loop does not bound any 2-chain in the MAI manifold at the working scale.

In summary, contractivity yields a unique fixed point and convergence; the polygonal chain of iterates closes in the limit to a 1-cycle $\gamma$ with $\partial\gamma = 0$, which is nontrivial whenever no 2-chain fills it. Hence structural recursion realizes topological closure as a latent recurrence. $\square$

## B  BIOLOGICAL IMPLEMENTATION OF TOPOLOGICAL CLOSURE

Oscillations discretize time on a circle ($S^1$), providing phase bins within which coincidence detection collapses fragments into recurrent traversals. Mathematically, the boundary calculus enforces this filtration: $\partial^2 = 0$ cancels unmatched endpoints so that only closed chains survive as persistent cycles. Cognitively, isolated tokens (dots) do not stabilize memory; only when linked by contextual relations into cycles do they consolidate as durable traces. In this section, we show how oscillatory phase coding and coincidence detection implement temporal scaffolding and boundary cancellation in spiking networks, turning temporal fragments into cycles.

## B.1 Oscillation Phase Coding as Temporal Scaffolding

Neural oscillations instantiate the closure principle by quotienting linear time to a circle: an oscillator implements $t \mapsto e^{i\omega t} \in S^1$, so events are registered by *phase* rather than absolute time. Biologically, this scaffold is realized at multiple, coupled timescales. (i) *Theta–gamma nesting* (e.g., hippocampus–entorhinal) provides a macrocycle ($\theta$, 4–12 Hz) that segments experience and a microcycle ($\gamma$, 30–100 Hz) that tiles each $\theta$ bin with ordered subevents; phase–amplitude coupling thus lays out a toroidal code $S^1_\theta \times S^1_\gamma$ in which winds index recurrent cycles Lisman & Jensen (2013); Canolty & Knight (2010); Canolty et al. (2006). (ii) *Coincidence detection* sharpens edges of these cycles: NMDA nonlinearity, backpropagating spikes, and fast interneuron circuitry (PV/ING, PING) create narrow $\mathcal{O}(1-10 \text{ ms})$ windows so that only spikes aligned within a phase bin form effective synaptic links; misaligned fragments fail to bind and are pruned König et al. (1996a); Stuart & Sakmann (1994); Buzsáki & Wang (2012). (iii) *Spike-timing dependent plasticity (STDP)* orients these links by phase lead/lag, turning phase offsets into directed edges in a chain; repeated traversal within a cycle consolidates these edges, canceling stray endpoints and favoring closed walks Markram et al. (1997); Bi & Poo (1998); Caporale & Dan (2008a). (iv) *Conduction delays and myelin plasticity* tune effective phase lags, enabling polychronous assemblies: axonal/dendritic delays align distributed spikes into reproducible phase patterns that complete cycles despite spatial dispersion Izhikevich (2006); Pajevic et al. (2014); Fields (2015). (v) *Phase-of-firing coding and precession* (e.g., hippocampal place cells) map position or task progress to phase on $S^1_\theta$, so that a behavioral episode corresponds to a return map on the Poincaré section; complete laps close in phase space, incomplete traversals do not O'Keefe & Recce (1993); Montemurro et al. (2008). (vi) *State-dependent reentry* (sharp-wave ripples during NREM/quiet wake) replays phase-ordered sequences on a faster carrier, tightening weights along already-closed paths and suppressing nonclosing detours Foster & Wilson (2006); Diba & Buzsáki (2007).

*Interpretation.* Oscillations supply the contextual scaffold $\Psi$ that folds timelines into cyclic coordinates; coincidence and plasticity then implement boundary cancellation in synaptic space. What persists are cycles, phase-locked traversals whose endpoints identify on $S^1$, while unmatched fragments dissipate. This sets up the formal lemma below, which recasts phase-binned spiking as a chain whose boundary vanishes after a full cycle.

**Lemma 3** (Oscillatory Phase Coding as Temporal Scaffolding). *Let $\theta(t) = \omega t \pmod{2\pi}$ denote the phase of a neural oscillator, with events encoded relative to $\theta(t)$ on the circle $S^1$. Then oscillatory phase coding induces the following invariants: 1)* **Binding:** *Events occurring within the same phase window $\theta(t) \in [\phi, \phi + \Delta]$ are grouped together, forming a coherent representation; 2)* **Ordering:** *Sequences of events are represented by their relative phase offsets $(\Delta\theta_1, \Delta\theta_2, \dots)$, embedding linear order into a cyclic scaffold; 3)* **Closure:** *After a full cycle $\theta(t + T) = \theta(t)$ with $T = \frac{2\pi}{\omega}$, the system resets, ensuring that trajectories are organized into cycles rather than unbounded chains. Together, these properties enforce the topological identity $\partial^2 = 0$ at the temporal level: the boundary of one temporal segment becomes the beginning of the next, so that each cycle closes before a new one begins. Consequently, oscillatory phase coding guarantees consistency of memory traces by embedding them in recurrent temporal cycles.*

The formal statement of Lemma 3 captures how oscillatory phase coding transforms linear time into a cyclic scaffold, guaranteeing binding, ordering, and closure. To visualize this principle, Fig. 4 illustrates how linear time $t$ is wrapped onto the circle $S^1$ (theta phase), with discrete gamma packets embedded at distinct phases. Events that fall into the same phase window (green arc) are bound together, while relative phase offsets encode ordering. The reset at the end of each $\theta$ cycle ensures closure, embodying the algebraic identity $\partial^2 = 0$ in biological timekeeping. Let $\theta : \mathbb{R} \to S^1$ be the phase map $\theta(t) = \omega t \bmod 2\pi$, so $T = \frac{2\pi}{\omega}$ identifies $t \sim t + T$ and quotients linear time to a circle. Partition $S^1$ into $L$ phase bins $\{\varphi_\ell\}_{\ell=1}^L$ and let $v_\ell$ denote the (phase-binned) latent state aggregated within bin $\varphi_\ell$. Define oriented edges $e_\ell = [v_\ell, v_{\ell+1}]$ with $v_{L+1} \equiv v_1$. The phase-ordered chain $c = \sum_{\ell=1}^L e_\ell$ has $\partial c = \sum_{\ell=1}^L (v_{\ell+1} - v_\ell) = v_{L+1} - v_1 = 0$, so a full $2\pi$ sweep closes into a 1–cycle. Coincidence detection enforces this construction: only events aligned within a phase window of width $\varepsilon$ create edges, pruning stray fragments whose endpoints would otherwise fail to cancel. Conduction delays implement modular jumps $e_\ell = [v_\ell, v_{\ell+k}]$, yielding a winding number $k$ on $S^1$; after $L$ such steps the path returns to $v_1$, again giving $\partial c = 0$ and a homology class $[c] \in H_1(S^1) \cong \mathbb{Z}$.

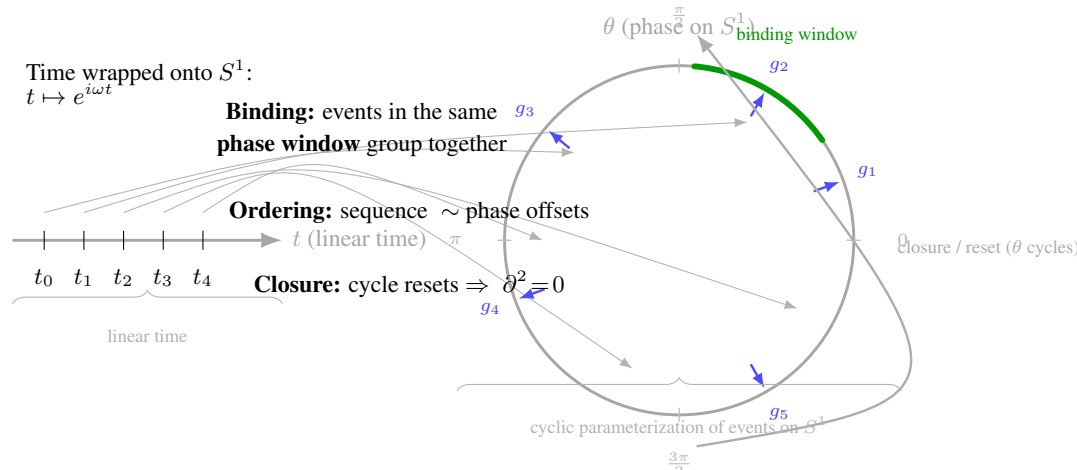

Figure 4: **Oscillatory phase coding as topological closure.** Linear time is wrapped onto the circle $S^1$ (theta phase), placing events by *phase* rather than absolute time. Gamma packets $(g_1, \ldots, g_5)$ at distinct phases encode *order* via phase offsets, while events within the same phase window (green arc) are *bound*. Cycle reset at the end of each theta period enforces topological closure ($\partial^2 = 0$), supporting consistent memory cycles.

Two useful views follow. (i) *Poincaré/return map:* sampling at phase $\phi_0$ defines $F_{\phi_0} : x(t) \mapsto x(t + T)$; fixed points and periodic points of $F_{\phi_0}$ are closed orbits on $S^1$, i.e., cycles Kolomiets & Shilnikov (2020). (ii) *Cross-frequency nesting:* with $\theta$ and $\gamma$ phases, time quotients to a torus $S^1_\theta \times S^1_\gamma$ and $H_1 \cong \mathbb{Z}^2$; winds $(k_\gamma, k_\theta)$ encode hierarchical cycles Belluscio et al. (2012). *Summary.* Phase coding turns linear sequences into cyclic invariants: the absolute start/end times are identified on $S^1$, so boundaries telescope away and only closed traversals persist. Coincidence gates which edges exist; $\partial^2 = 0$ guarantees unmatched endpoints cannot accumulate into memory, while completed cycles survive as stable traces.

## B.2 COINCIDENCE DETECTION AS TOPOLOGICAL CLOSURE

Lemma 3 establishes that oscillatory phase coding furnishes a natural scaffold for aligning events on the circle $S^1$, ensuring that candidate trajectories can be organized into cyclic frames. However, phase alignment alone does not guarantee stability: without a mechanism to prune misaligned or inconsistent events, spurious boundaries would accumulate and prevent reliable cycle formation. Lemma 4 addresses this gap by showing how coincidence detection enforces closure at the level of spike trains, cancelling mismatches as boundary terms and preserving only those cycles that survive across trials. Together, these results formalize the complementary roles of phase scaffolding and coincidence pruning in transforming transient alignments into reproducible cognitive invariants.

**Lemma 4** (Coincidence-Induced Closure and Survival of Reproducible Cycles). *Let $\mathcal{N} = \{1, \ldots, n\}$ be a set of units (neurons) producing a spike train $S = \{(i, t_k)\}$ with phases $\phi(t_k) \in S^1$. Fix a coincidence window $\Delta \in (0, \pi)$ and define the* coincidence relation $i \overset{\Delta}{\leftrightarrow} j$ *iff there exist spikes $(i, t), (j, t') \in S$ with $|\phi(t) - \phi(t')|_{S^1} \leq \Delta$ and $t < t'$ (to orient time). Construct the directed 1–skeleton $G_\Delta(S)$ whose vertex set is $\mathcal{N}$ and whose (possibly multiple) oriented edges are $e = (i \to j)$ for every coincident pair $i \overset{\Delta}{\leftrightarrow} j$. Let $C_1(G_\Delta)$ be the free abelian group on edges and $C_0(G_\Delta)$ the free abelian group on vertices, with boundary $\partial : C_1 \to C_0$ given by $\partial(i \to j) = j - i$. Define the* coincidence aggregation $c_\Delta(S) \in C_1(G_\Delta)$ *by summing all oriented edges (with multiplicities) generated by coincident pairs, and the* coincidence projection $\Pi_\Delta : C_1(G_\Delta) \longrightarrow Z_1(G_\Delta) := \ker \partial$ *as the (linear) projection onto the cycle space (e.g., orthogonal projection with respect to any inner product on $C_\bullet$ or the canonical decomposition $C_1 = Z_1 \oplus B_1^\perp$). Then we have: 1) **Closure by coincidence.** The* coincidence detector $K_\Delta := \Pi_\Delta \circ (\cdot)$ *enforces closure: $z_\Delta(S) := K_\Delta(c_\Delta(S)) \in Z_1(G_\Delta)$ and $\partial z_\Delta(S) = 0$. Moreover, the edges removed by $K_\Delta$ are precisely those whose net contribution appears in $\partial c_\Delta(S)$; i.e.*

*misaligned spikes are canceled as boundary terms and do not survive in $z_\Delta(S)$. 2) **Survival of reproducible cycles (stability).** Suppose $S^{(1)}, \ldots, S^{(T)}$ are trials with phase jitter at most $\varepsilon < \Delta$ (i.e. every coincidence in one trial has a matched coincidence within phase distance $\varepsilon$ in all others, with the same orientation). Then for all $t$, $[z_\Delta(S^{(t)})] = [z_\Delta(S^{(1)})] \in H_1(G_\Delta; \mathbb{Z})$, so the homology class is trial-invariant. In particular, in the persistence module obtained by varying the window $\delta \in (\varepsilon, \Delta]$, this class has positive lifetime and therefore* survives *while nonreproducible coincidences die as boundaries.*

*Proof sketch.* (1) By construction, $\partial c_\Delta(S)$ counts net imbalance of incident coincidences at each vertex (incoming minus outgoing). Projecting onto $\ker \partial$ removes exactly those components whose boundary is nonzero; hence $z_\Delta(S) \in Z_1$ and $\partial z_\Delta(S) = 0$. Informally, coincidences that do not close are eliminated as boundary terms; only closed flow persists. (2) Phase jitter $\varepsilon < \Delta$ induces edge correspondences between the $G_\Delta(S^{(t)})$ that preserve orientation and incidence, yielding chain homotopic $c_\Delta(S^{(t)})$. Projection to $Z_1$ commutes with these homotopies, so the resulting $z_\Delta(S^{(t)})$ are homologous. Viewing $\delta$ as a filtration parameter, unmatched (nonreproducible) edges vanish at $\delta \searrow \varepsilon$, whereas reproducible cycles define a bar of positive length in $H_1$, hence survive. $\qquad\square$

Lemma 4 established that coincidence detection enforces closure by cancelling misaligned spikes, ensuring that only reproducible cycles survive. Fig. 5 illustrates this principle: when presynaptic spikes align within a coincidence window $\Delta$ (top), their inputs sum coherently and trigger a postsynaptic spike, corresponding to $\partial\gamma = 0$ (closure). When spikes fall outside the window (bottom), they remain as unmatched boundaries that cancel one another, yielding no output. The inset shows the topological analogy: different paths that bound the same face $\sigma$ cancel in homology, just as misaligned temporal fragments fail to stabilize into persistent cycles. Formally, we have

**Definition 3** (Topological Closure). *Let $(X, \tau)$ be a topological space and $A \subseteq X$. The* closure *of $A$, denoted $\overline{A}$, is defined as $\overline{A} = \bigcap\{C \subseteq X \mid C \text{ is closed and } A \subseteq C\}$. Equivalently, $\overline{A}$ consists of all points $x \in X$ such that every open neighborhood $U \in \tau$ with $x \in U$ satisfies $U \cap A \neq \varnothing$.*

With the formal notion of closure in hand, we now *operationalize* it in neural dynamics: replace open neighborhoods by temporal coincidence windows and subsets $A$ by sets of candidate spikes. Under this identification, the "points in the closure" are precisely spikes that recurrently co-occur within a window, and the homological reading of closure ($\partial^2 = 0$) corresponds to cancelling unmatched, out-of-window events as boundary terms. This yields a direct bridge from topological closure to coincidence-driven cycle formation in neural circuits. For a PNG to persist, spikes from multiple presynaptic neurons must converge within a narrow temporal window at their postsynaptic targets. Coincidence detection acts as a filter: inputs that arrive in synchrony are integrated, while those that fall outside the coincidence window are effectively cancelled. This selective integration implements the algebraic identity $\partial^2 = 0$: misaligned spikes behave like open boundaries that fail to connect, whereas synchronous arrivals cancel boundary terms and enforce cycle closure. In this way, only temporally coherent activity contributes to a closed 1-cycle in the neural state space. Once closure is achieved, spike-timing dependent plasticity (STDP) reinforces the recurrent pathways that produced coincident input Caporale & Dan (2008b). Potentiation strengthens the synapses along routes that consistently deliver spikes within the window $\Delta$, while depression weakens those that fail to align. Over repeated activations, this differential plasticity stabilizes the trajectory as a reentrant cycle: the cycle not only replays reliably, but also becomes resistant to perturbations of individual spike times. In summary, coincidence detection, together with STDP, extracts the low-entropy content variable $\Phi$: a reproducible invariant that persists as a memory trace and can later be recalled or recombined into higher-order structures Li (2025c).

The principle "coincidence detection = boundary cancellation" can now be made explicit. When presynaptic spikes converge within the coincidence window, their temporal boundaries align and cancel, producing a closed cycle that can drive a stable postsynaptic response König et al. (1996b). In contrast, when spikes arrive outside the window, they leave residual unmatched boundaries that fail to close, and no postsynaptic output is generated. Figure 5 illustrates this correspondence: in the neural case, misaligned spikes cancel each other's contributions and disappear from the effective cycle; in the topological case, paths that differ by the boundary of a 2-simplex $\sigma$ cancel in homology. In both settings, coincidence detection enforces the identity $\partial^2 = 0$, ensuring that only closed cycles survive as memory-bearing invariants.

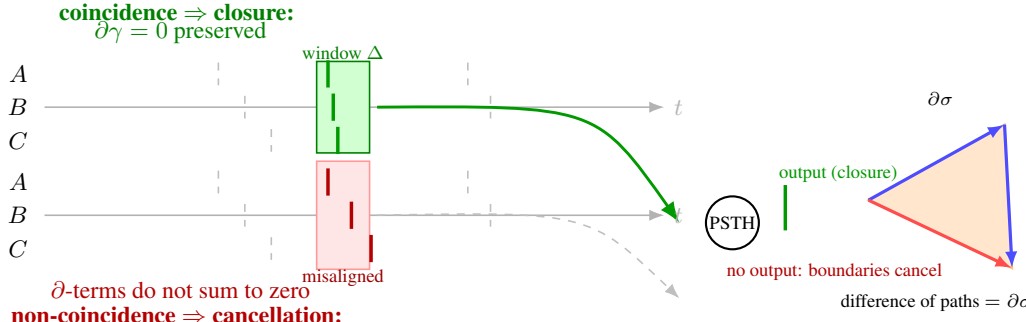

Figure 5: **Coincidence detection = boundary cancellation.** *Left:* Three presynaptic spike trains $(A, B, C)$. **Top:** Spikes align within a coincidence window $\Delta$ (green band), summate, and produce a postsynaptic spike (closure). **Bottom:** Spikes are misaligned; inputs fail to coincide in $\Delta$, so no output occurs (boundaries do not cancel). *Right inset:* Topological analogy - when two paths differ by the boundary of a face ($\partial\sigma$), their difference cancels in homology; likewise, misaligned temporal fragments behave as open boundaries, while coincidence implements $\partial^2 = 0$, leaving only closed cycles.

**Principle 4** (Coincidence Detection as Boundary Cancellation). *Let $\{v_i\}$ denote neural events (e.g., spikes) indexed in time, and let $e_{ij} = [v_i, v_j]$ denote a directed edge formed when two events fall within a coincidence window $\Delta t$. Define a chain $c = \sum_{(i,j)} e_{ij}$ over all coincident pairs. 1) If events are misaligned ($|t_i - t_j| > \Delta t$), no edge is formed; the fragment remains an open chain with nonvanishing boundary $\partial e_{ij} = v_j - v_i$. 2) If events are coincident ($|t_i - t_j| \leq \Delta t$), opposite boundaries cancel: $\partial c = \sum(v_j - v_i) = 0$. In summary, coincidence detection implements the algebraic rule $\partial^2 = 0$: unmatched endpoints dissipate, while synchronous inputs enforce closure. Biologically, this ensures that only temporally aligned inputs reinforce into stable cycles, whereas misaligned fragments are pruned.*

## C    MAI AS TIME-REVERSED REINFORCEMENT LEARNING

Non-ergodicity offers a principled foundation for understanding both reinforcement learning (RL) and its time-reversed dual, memory-amortized inference (MAI). In RL, the agent iteratively descends through state-action trajectories to minimize expected future cost via bootstrapped value updates Schultz et al. (1997). This process inherently assumes a forward temporal flow, where actions alter state and reward accumulates over time. However, non-ergodic agents do not uniformly explore the state space; rather, they converge onto structured attractors, recurrent paths, policies, or goals, due to the reuse of historical structure. Therefore, it is non-ergodicity that ensures important states recur as broken symmetry Anderson (1972), allowing the RL system to build and refine value estimates via temporal bootstrapping.

MAI formalizes this broken symmetry in reverse: instead of descending value gradients to reach future states, it reuses predicted future states (e.g., $\Phi_{t+1}$) to retrieve prior memory states ($\hat{\Phi}_t$) consistent with the current context. As time-reversed bootstrapping, MAI performs inference not by forward reward accumulation, but by backward alignment with structured latent cycles. Both processes are constrained by persistent topological features (e.g., homology classes, attractor submanifolds), but differ in directionality: RL propagates utility forward; MAI propagates structure backward. This duality reveals that *non-ergodicity not only explains the emergence of RL*, but also necessitates a complementary *reverse-time inference mechanism*, captured by MAI, to efficiently simulate, adapt, and generalize in structured cognitive systems. This section formalizes the duality between RL and MAI under a time-reversal transformation, revealing deep structural parallels between bootstrapped value updates and latent cycle inference.

**Time-Reversal Duality Between RL and MAI.** Let $V(s_t)$ denote the value function at state $s_t$. The temporal-difference (TD) update rule is Sutton & Barto (1998): $V(s_t) \leftarrow V(s_t) + \alpha\left(r_t + \gamma V(s_{t+1}) - V(s_t)\right)$ This rule bootstraps the estimate of $V(s_t)$ from the next state's value

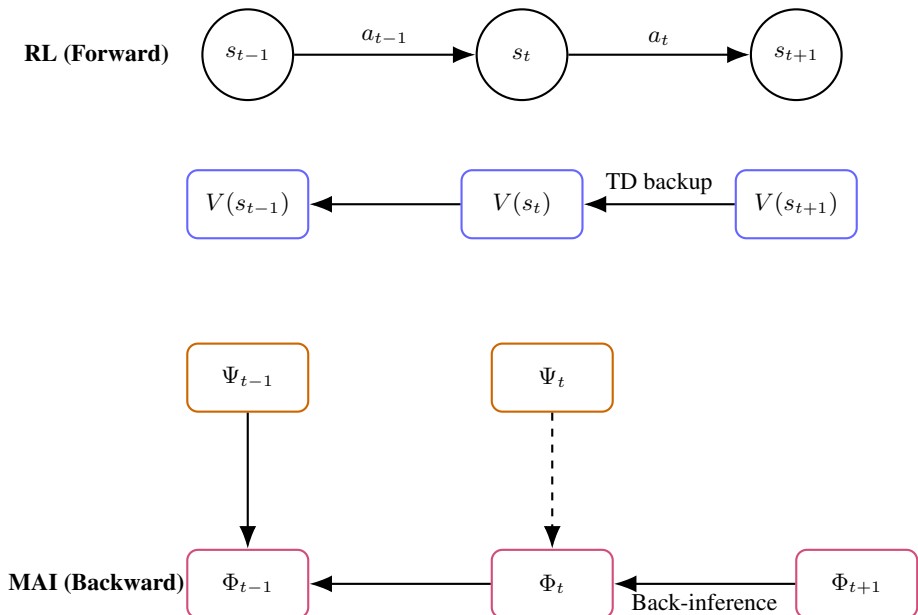

Figure 6: Duality between Reinforcement Learning and Memory-Amortized Inference. RL proceeds forward in time using TD backup; MAI proceeds backward using context-guided reconstruction over latent memory cycles.

$V(s_{t+1})$, defining a one-step look-ahead propagation. More generally, the Bellman expectation equation governs the forward dynamics: $V(s_t) = \mathbb{E}_{a_t, s_{t+1}} [r_t + \gamma V(s_{t+1})]$ In MAI, the system simulates past latent content $\Phi_{t-1}$ from current content $\Phi_t$ and context $\Psi_{t-1}$. The inference operator takes the form: $\Phi_{t-1} = \mathcal{R}(\Phi_t, \Psi_{t-1})$, $\Phi_t = \mathcal{F}(\Phi_{t-1}, \Psi_{t-1})$ MAI thus implements a *backward cycle-consistent process* Li (2025b), amortizing reconstruction of prior states via topological reuse of structured latent memory. Together, RL and MAI break the separation between learning and inference by forming a closed loop. An important new insight brought by such time-asymmetric bootstrapping is "plan by bootstrapping forward; infer by bootstrapping backward", which might unlock the secret for enabling low-energy, high-efficiency cognition. The time-reversal duality between RL and MAI is summarized in Table 1 below.

| Dimension | Reinforcement Learning (RL) | Memory-Amortized Inference (MAI) |
|---|---|---|
| Time direction | Forward (future-oriented) | Backward (past-inference) |
| Bootstraps | Expected future rewards | Latent priors from future predictions |
| Undertainty type | *Outcome uncertainty* (What will happen?) | *Inference uncertainty* (What generated this?) |
| Policy element | $\pi(a_t\|s_t)$, learned from reward | $\Psi_t$, learned from structure-consistency |
| Bias source | Reward shaping, value iteration | Latent memory structure, trajectory reuse |
| Learning type | Goal-directed exploration | Context-conditioned generalization |
| Half-step trick | TD: $V(s_t) \approx V(s_{t+1})$ | MAI: $\Phi_t \approx \mathcal{R}(\Phi_{t+1}, \Psi_t)$ |

Table 1: Time-Reversed Duality between Reinforcement Learning and Memory-Amortized Inference

Table 1 illustrates a fundamental time-asymmetric duality between RL and MAI. Whereas RL operates forward in time, projecting expected rewards to guide future actions via temporal difference (TD) updates, MAI runs in reverse, retrieving latent priors from predicted futures to reconstruct past inference trajectories Sutton & Barto (1998), as shown in Fig. 6. This duality is not superficial; it reflects a structural inversion of uncertainty management: RL reduces *outcome uncertainty* through forward value propagation, while MAI minimizes *inference uncertainty* by aligning predictions with memory cycles. Both leverage bootstrapping to avoid recomputation, yet from opposite directions: RL refines estimates via anticipated value; MAI refines inference via recovered structure. The "half-step trick" in each case captures this temporal asymmetry Watkins & Dayan (1992): RL assumes $V(s_t) \approx V(s_{t+1})$, while MAI assumes $\Phi_t \approx \mathcal{R}(\Phi_{t+1}, \Psi_t)$. This symmetry-breaking across time

motivates a deeper unification under the Entropy-Reversibility Duality, wherein inference and behavior emerge from structurally consistent, direction-sensitive updates over latent space.

**From Time-Reversal to Entropy-Reversibility Duality.** The time-reversal duality between RL and MAI reveals a deeper symmetry: both are bootstrapping mechanisms that operate in opposite temporal directions to manage uncertainty. RL projects value forward to guide action, while MAI reuses future predictions to reconstruct latent causes. Despite this temporal asymmetry, both processes exploit structural regularities (e.g., recurring states, cycles, attractors) to reduce computational costs and enhance generalization.

This motivates a broader unifying principle: the *Entropy-Reversibility Duality*. Under this principle, forward processes like RL reduce outcome uncertainty through reversible value propagation, whereas reverse-time processes like MAI reduce inference uncertainty through structural reversibility over latent trajectories. Intelligence, then, is not merely the product of forward planning or backward recall, but of a *cycle-consistent interplay* between reversible inference and entropy minimization across time. This duality suggests that efficient cognition arises when entropy is tamed by structural reversibility, where memory not only stores outcomes but constrains their generative causes. In this framework, the direction of time aligns with the direction of entropy flow: 1) In RL, entropy is reduced by selecting high-value actions from many possible futures; 2) In MAI, entropy is minimized by reconstructing low-entropy past content from high-entropy contextual traces. This suggests that reversible inference is possible only when structural entropy is preserved and amortized through cyclic reuse. Formally, we have

**Theorem 4** (Entropy–Reversibility Duality). *Let $\mathcal{S} = \{s_t\}$ be a forward-time trajectory under a reinforcement learning policy $\pi$, and let $\mathcal{M} = \{\Phi_t, \Psi_t\}$ be a backward-time latent memory trajectory under MAI. Suppose: 1) The entropy rate of the forward trajectory satisfies $H(\mathcal{S}) = H[s_t|s_{t-1}]$; 2) The amortized inference process satisfies cycle-consistency: $\Phi_t \approx \mathcal{R}(\mathcal{F}(\Phi_t, \Psi_t), \Psi_t)$; 3) The joint entropy of memory satisfies $H(\Phi_t, \Psi_t) < H(\Psi_t) + H(\Phi_t)$ (i.e., structural dependence). Then the following duality holds: Minimizing entropy in MAI $\iff$ Reversing value propagation in RL Moreover, reversible inference is possible if and only if the entropy difference $\Delta H = H(s_{t+1}) - H(\Phi_{t-1})$ is bounded by the amortized structural information reused across the cycle.*

*Sketch.* RL reduces entropy by forward compression: selecting actions reduces future uncertainty. MAI reduces entropy by backward reconstruction: reusing structured cycles limits the degrees of freedom needed to infer latent causes. When the memory space encodes sufficient redundancy, reversing inference becomes possible under bounded entropy. The equivalence follows from the conservation of uncertainty across the reversed Markov chain induced by memory cycles. $\square$

**Implications for Learning Systems.** Theorem 4 articulates a unifying constraint on intelligent behavior: the capacity to minimize entropy, whether in inference or control, depends critically on the reversibility of internal computation. This has several far-reaching implications for the design and understanding of both biological and artificial learning systems:

- **Reversible inference is a structural necessity.** It is not merely a computational shortcut or an architectural convenience, but a reflection of an underlying physical and information-theoretic law: entropy reduction requires structural recurrence. Inference processes that do not re-enter prior states or cycles are fundamentally limited in their ability to generalize or compress. Hence, intelligent systems must harness structural memory, latent attractors, or topological cycles to sustain low-entropy prediction.

- **Model-based RL approximates MAI through reuse.** RL agents that construct internal models of state dynamics increasingly resemble memory-amortized systems. When these agents simulate or plan using cached transitions, they implicitly rely on structure-preserving reuse, thus operating within the MAI regime. This suggests that the historical separation between planning and inference may be artificial, as both emerge from reversible, memory-centered architectures.

- **General intelligence emerges near the reversibility threshold.** Whether in the brain or artificial agents, the hallmark of general intelligence is not brute-force exploration or statistical averaging, but the ability to cyclically refine predictions and actions using compact, reusable representations. This occurs near a critical point, which we term the *reversibility*

*threshold*, where entropy reduction and structural conservation are co-optimized. Systems operating at this threshold exhibit high energy efficiency, fast adaptation, and robustness to uncertainty, suggesting it may be a necessary condition for superintelligent cognition.

**Example 3** (Reversibility Threshold in Route Replanning). *Consider an agent navigating a spatial environment (e.g., a robot exploring a maze or a human traversing a familiar city). Initially, the agent explores via stochastic policies, with high entropy over possible trajectories due to a lack of structural knowledge. As it accumulates experience, it begins to form latent memory cycles that encode recurrent paths (e.g., loops through landmarks). When a roadblock appears, the agent does not re-explore from scratch; instead, it performs a localized inference over prior trajectories, retrieving structurally similar detours from memory. At this stage, the agent operates near the* reversibility threshold*: its predictions and actions are no longer purely exploratory (high entropy), nor fully deterministic (low flexibility). Instead, the system achieves a critical balance where entropy is minimized through reuse of past cycles (low-entropy inference). Meantime, structural representations (e.g., topological homology classes over trajectories) are preserved across updates; and adaptation remains possible through bootstrapped generalization of known paths. This regime exemplifies co-optimization of entropy and structure: inference proceeds efficiently with minimal recomputation, while remaining reversible through cyclic memory access. If structural reuse were impaired (e.g., due to memory corruption), the agent would regress to high-entropy re-exploration. Conversely, if flexibility were lost (e.g., overfit to a single path), the agent could not adapt. Thus, intelligence manifests most effectively at this* reversibility threshold.

RL models reward uncertainty by estimating the expected return over stochastic transitions and actions, i.e., $\mathbb{E}[r]$, which underpins policy optimization via value iteration or temporal difference learning Sutton & Barto (1998). However, such formulations assume that reward signals are available and semantically meaningful at each state, and that learning progresses through direct interaction with the environment. MAI offers a principled generalization: rather than modeling uncertainty solely over scalar rewards, MAI models the uncertainty over latent causes of observed outcomes, conditioned on context $\Psi$. Specifically, MAI replaces the simple expected reward $\mathbb{E}[r]$ with a nested expectation $\mathbb{E}_{\Psi}[\mathbb{E}[r|\Psi]]$, where the inner expectation is over memory-retrieved experiences structurally consistent with $\Psi$. This shift enables inference over abstract, context-sensitive value functions even in the absence of immediate feedback, unifying reward estimation with memory reuse and structural generalization under a single retrieval-and-adaptation framework. Formally, we have

**Theorem 5** (Contextual Expectation in Memory-Amortized Inference). *Let $\mathcal{D}_{\Psi}$ be a distribution over contexts $\Psi \in \mathcal{X}_{\Psi}$, and suppose there exists a latent memory store $\mathcal{M} = \{(\Psi^{(i)}, \Phi^{(i)}, r^{(i)})\}_{i=1}^{N}$, where each tuple stores context, content, and a reward-like utility signal. Let $\mathcal{R} : \mathcal{X}_{\Psi} \times \mathcal{M} \to \mathcal{X}_{\Phi}$ be a retrieval operator such that $\hat{\Phi}(\Psi) := \mathcal{R}(\Psi; \mathcal{M}) \approx \mathbb{E}_{\Phi^{(i)} \sim \mathcal{M}|\Psi}[\Phi^{(i)}]$. Then the expected utility of amortized inference is given by a doubly nested expectation: $\mathbb{E}_{\Psi \sim \mathcal{D}_{\Psi}}[\mathbb{E}_{\Phi^{(i)} \sim \mathcal{M}|\Psi}[r^{(i)}]]$, which generalizes reinforcement learning's expected reward: $\mathbb{E}[r] \rightsquigarrow \mathbb{E}_{\Psi}[\mathbb{E}[r|\Psi]]$. This formulation enables structure-aware generalization via context-conditioned reuse of past experience.*

# D    STATEMENT ON THE USE OF LARGE LANGUAGE MODEL (LLM)

The author has utilized ChatGPT models (Auto Mode of V5) to aid in the development of theoretical ideas, the proof of theorems/lemmas/propositions, as well as the visual illustrations presented in this paper. However, the author is responsible for all key ideas (through providing prompts and iterative Q&As), the overall paper organization, the logical flow within each section, and the polishing of the paper draft (e.g., to eliminate glaring artifacts such as dashes or "—" generated by the LLM) and all figures (e.g., resize the figure size and adjustment of inserted texts and captions).

