# OpenReview forum: "Beyond Turing: Topological Closure as a Foundation for Cognitive Computation"
_ICLR.cc/2026/Conference — ICLR 2026 Conference Withdrawn Submission_

### Official Review · Reviewer_eTHf · 2025-10-23

**Soundness:** 1
**Presentation:** 1
**Contribution:** 1
**Rating:** 0
**Confidence:** 2

**Summary:**

The paper claims to propose a new theoretical framework for understanding computation and cognition beyond the classical Turing paradigm. It introduces several high-level principles about cognition and perception, followed by a shift into a purportedly mathematical formalization involving concepts such as “boundary operators” and “topological transitions.”

**Strengths:**

- The paper is ambitious in scope, attempting to bridge cognitive science, philosophy of computation, and mathematical modeling.

- The motivation — rethinking computation in light of cognitive dynamics — is potentially interesting and aligns with interdisciplinary goals.

**Weaknesses:**

The paper is extremely difficult to follow. The exposition mixes vague cognitive slogans with partially formalized mathematics, without properly defining key terms or notations (e.g., “boundary operator” appears without prior introduction).

Theorems and claims are presented in a hybrid style that combines mathematical language with informal cognitive interpretations, making it unclear whether they are meant as formal results or conceptual metaphors.

The conceptual framing of cognition and computation is very general and abstract, with little empirical or computational grounding.

The paper’s style and content will likely be intelligible only to a small subset of readers already sharing the authors’ conceptual framework; for the broader ICLR audience, it lacks clarity, accessibility, and demonstrable contribution.

**Questions:**

1. Could you clarify how you see this work fitting within the ICLR community? The paper’s focus appears primarily conceptual and philosophical, with limited computational or empirical grounding. It is not clear whether the intended contribution is theoretical (mathematical), cognitive-scientific, or computational.

2. Do you envision any practical implications or experimental predictions that could make the framework relevant to machine learning or neural computation?

3. Would a different venue—for instance, one oriented toward philosophy of computation or mathematical cognition—better suit the goals of this work?

---

### Official Review · Reviewer_vVJR · 2025-10-28

**Soundness:** 1
**Presentation:** 1
**Contribution:** 1
**Rating:** 0
**Confidence:** 4

**Summary:**

Authors try to propose that topological closure is a unified framework for cognitive computation that is better than Turing machine.

**Strengths:**

The authors claim that the cycle is all you need.

**Weaknesses:**

This paper very hard to read.

The current submitted draft is a mishmash and contains many misconceptions and inconsistencies, e.g., “computation is grounded in enumeration”, “computation is the syntactic manipulation of discrete symbols”.

 “Deep learning architectures … inherit enumerative character: generalization emerges by statistical interpolation over enumerated training examples” – it is still not clear whether deep learning systems can indeed perform generalization. Gödel’s incompleteness theorem is only about incompleteness.

 Memory is better understood as the ability to re-enter and traverse latent cycles in the neural state space. I understand there, memory simulates the working-memory, not the long-term memory.

“Cognition is not tape-based symbol manipulation but the promotion of transient fragments into closed cycles” – Symbol manipulation is the syntactic operations to realise the state-transition within the semantic world. The authors completely neglected the semantics behind the syntactic operations.

“To operationalize this picture in cognition, we adopt the Context–Content Uncertainty Principle
(CCUP) Li (2025a)”. This reference is noted as “under review”.

All \cite{} shall be replaced with \citep{}.

**Questions:**

Can all syntactic manipulation be reduced to enumeration? What is “enumeration”?  “enumeration can never guarantee closure” What is “closure”? How is it related with completeness? What are “Residual boundaries” and “open fragments”?  What failures are called “distributional failures”? What is “topological closure”?

Philosophically, if we distinguish fiat boundary from bona fide boundary, can your boundary operator be applied for both?

---

### Official Review · Reviewer_WHiK · 2025-11-01

**Soundness:** 1
**Presentation:** 1
**Contribution:** 2
**Rating:** 0
**Confidence:** 3

**Summary:**

This paper argues against the classical Turing model of computation, and instead argues for an alternative foundation for cognition based on the notion of "topological closure", which is the idea that cycles rather than symbolic enumeration better capture the nature of memory, reasoning and intelligence. Using concepts from algebraic topology (such as homology and boundary operators), the authors develop a metaphorical and mathematical framing outlining this idea.

**Strengths:**

The paper raises potentially thought-provoking questions about the structural foundations of cognition.

**Weaknesses:**

This paper does not seem like the best fit for ICLR. It doesn't include any machine learning experiments or any theoretical treatments of machine learning systems.

1. The paper is much too abstract and speculative. The text reads more like a philosophical essay.

2. There is no formal learning problem or setting to say anything concrete. The theorems and proofs  (while I'm not sure whether they are correct) study abstract mathematical objects without instantiating any particular learning setting. It is unclear how to operationalize the definitions and results into something for machine learning. No empirical content or actionable theoretical proposal.

4. The paper doesn't engage with machine learning literature in any meaningful way.

5. There is no discussion about how to apply the framework or any consideration of its limitations.

**Questions:**

Questions:
- Is this a philosophical or position paper? If so, it may be more relevant in venues that accept position papers and also consider more foundational questions on cognition.
- If the goal is to influence machine learning thinking, can you make clearer connections to current practice or identify concrete research directions? Could the authors clarify the connection between their framework and current machine learning practice or research?
- What would be a minimal working example that demonstrates the practical relevance of your framework to machine learning?

---

### Note · Authors · 2025-11-12

I have read and agree with the venue's withdrawal policy on behalf of myself and my co-authors.